

# Personalized analysis of breast cancer using sample-specific networks

Ke Zhu[1,*], Cong Pian[1,*], Qiong Xiang[1], Xin Liu[1] and Yuanyuan Chen[1,2]

[1] College of Science, Nanjing Agricultural University, Nanjing, Jiangsu, China
[2] State Key Laboratory of Bioelectronics, School of Biological Science and Medical Engineering, Southeast University, Nanjing, Jiangsu, China
* These authors contributed equally to this work.

## ABSTRACT

Breast cancer is a disease with high heterogeneity. Cancer is not usually caused by a single gene, but by multiple genes and their interactions with others and surroundings. Estimating breast cancer-specific gene–gene interaction networks is critical to elucidate the mechanisms of breast cancer from a biological network perspective. In this study, sample-specific gene–gene interaction networks of breast cancer samples were established by using a sample-specific network analysis method based on gene expression profiles. Then, gene–gene interaction networks and pathways related to breast cancer and its subtypes and stages were further identified. The similarity and difference among these subtype-related (and stage-related) networks and pathways were studied, which showed highly specific for subtype Basal-like and Stages IV and V. Finally, gene pairwise interactions associated with breast cancer prognosis were identified by a Cox proportional hazards regression model, and a risk prediction model based on the gene pairs was established, which also performed very well on an independent validation data set. This work will help us to better understand the mechanism underlying the occurrence of breast cancer from the sample-specific network perspective.

## INTRODUCTION

According to the latest data from the survey of the International Agency for Research on Cancer (IARC) in 2018, the incidence of breast cancer is 24.2% among women worldwide, ranking first in female cancers (*Bray et al., 2018*). At present, the incidence of breast cancer is the highest, and its mortality ranks fourth in China. Breast cancer has strong heterogeneity. Based on the TNM staging system, breast cancer can be divided into Stages I, II, III, IV and V. There are many clinical types of breast cancer according to pathological classification and molecular classification. The pathological classification generally divides breast cancer into invasive and non-invasive breast cancer. The gold standard for the molecular typing of breast cancer is PAM50 molecular typing based on the expression profile of 50 genes, which classifies breast cancer into the Normal-like, LuminalA, LuminalB, Basal-like, and Her2 subtypes (*Perou et al., 2000*).

Corresponding author
Yuanyuan Chen,
chenyuanyuan@njau.edu.cn

The molecular typing of breast cancer has important reference value for clinical treatment of breast cancer. However, molecular typing requires transcriptome sequencing which is difficult to promote clinically. Currently, the diagnosis of breast cancer classification is mainly through immunohistochemistry (IHC), namely, diagnosis by the expression of four markers, ER (oestrogen receptor), PR (progestin receptor), HER2 gene (human epidermal growth factor receptor 2) and Ki-67 protein (proliferating cell nuclear antigen). ER and PR are important indicators for endocrine therapy and prognosis evaluation in breast cancer. Studies have shown that their expressions are positively correlated with total survival, treatment failure time, endocrine therapy response time, and recurrence time (*Hammond et al., 2010*; *Fitzgibbons et al., 2010*). In 2009, Cheang used GEP (gene expression analysis) to determine 14% as the threshold of Ki-67, which could be used to divide patients into two groups with good and bad prognoses (*Cheang et al., 2009*). In 2011, the St. Gallen International Expert Consensus agreed to include Ki-67 as an important standard for molecular typing, which is the key to distinguishing the Luminal A and Luminal B subtypes (*Goldhirsch et al., 2011*). In the growth and metastasis of breast cancer, HER2 is one of the most important factors, and its status can be used to predict the effect of drug treatment for breast cancer. Early detection and diagnosis and timely treatment are of great significance to improve the survival rate of breast cancer patients.

The etiology of breast cancer is still not clear, and there are many related factors, such as individual differences and a lack of effective treatments. With the development of biomedicine, personalized medicine is becoming the direction of breast cancer treatment in the future. At present, the medical plan can only be formulated through the study of single gene expression and mutation information. However, this information cannot fully reflect the personalized interaction and regulation among genes. Because onset and progression of cancer are often caused by the disruption of important biological networks such as cell cycle and apoptosis, but not a single gene. Indeed, there is a new and cutting-edge field of medical research, called network medicine, whose basic idea is that human diseases are rarely caused by single molecular determinant, but more likely influenced by a network of interacting molecular determinants with the propensity to cluster together in the human interactome (*Barabási, Gulbahce & Loscalzo, 2011*; *Conte et al., 2019*; *Fiscon et al., 2018*). Gene–gene interaction networks can reveal the interaction relations and regulatory mechanisms among genes, and they have the irreplaceable function of the single-gene monitoring of information (such as expression and mutation) in many aspects (*Liu et al., 2016*). Therefore, the mechanism of the occurrence and development of breast cancer can be explored through changes in the interactions between genes. In this article, we constructed sample-specific networks of breast cancer samples by calculating the correlation coefficient of protein-coding gene pairs to explore the gene–gene interaction networks related to breast cancer stages and subtypes (see Fig. 1).

The survival time of different patients with breast cancer is significantly different. At present, the 5-year survival rate of breast cancer patients in China has reached 83.2%. However, the 5-year survival rate of advanced cancer patients and Basal-like cancer

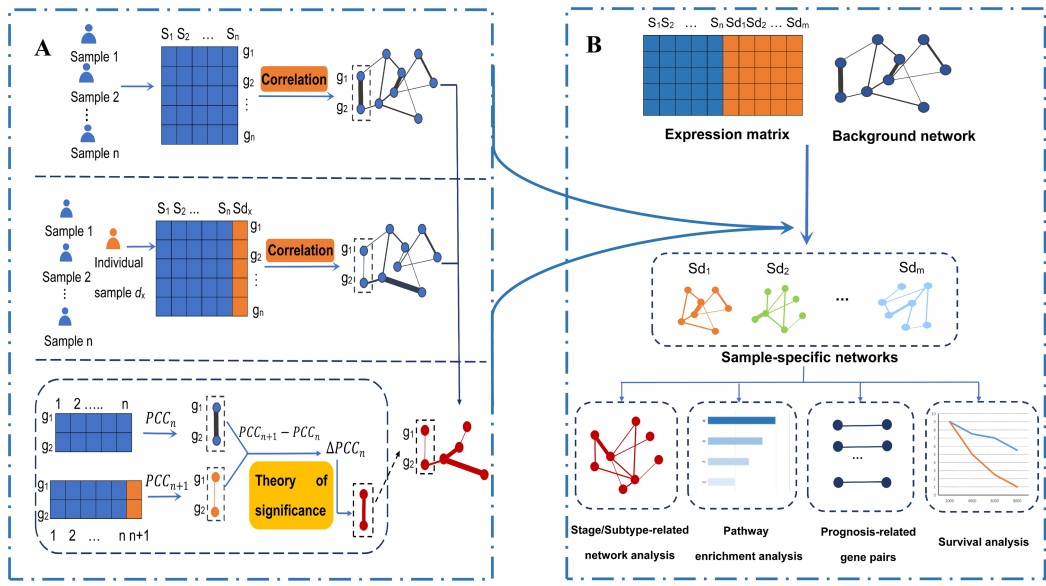

**Figure 1 An integrative framework identifying breast cancer-related gene–gene interaction networks.** (A) Construction of sample-specific networks based on gene expression data. A reference network can be established based on the expression profiles of $n$ reference samples by calculating the correlation coefficients $PCC_n$ of gene pairs. Then, adding a new cancer sample $sd_x$ into the reference samples, a perturbed network is established by calculating the new correlation $PCC_{n+1}$ of the $n + 1$ samples. Because of sample $sd_x$, the perturbed network is different from the reference network, and the difference $\Delta PCC_n$ ($PCC_{n+1} - PCC_n$) of each edge in the background network constitutes the differential network. Then, the significance of each edge can be quantified by a statistical $Z$-test. The sample-specific network for sample $sd_x$ is composed by those edges with significant $\Delta PCC_n$. (B) The framework to identify the breast cancer-related gene–gene interaction network based on gene expression. Using the sample-specific network analysis method, $m$ cancer sample-specific networks were constructed. Then, these constructed sample-specific networks were analyzed to identify breast cancer-related networks, stage-related networks and subtype-related networks, as well as gene-interaction biomarkers associated with the prognosis of breast cancer. Moreover, pathway enrichment analysis based on KEGG pathways and survival analysis based on the LASSO regression model were performed.

patients are significantly lower, so it is necessary to study the biomarkers that affect the prognosis of breast cancer. In 2009, *Parker et al. (2009)* established a single-gene level survival analysis model to improve the prognosis of breast cancer and predict the efficacy of chemotherapy. However, the robustness of the gene-based model is not very high. Thus, this paper aims to establish a more stable prognostic analysis model of breast cancer patients through gene–gene interactions. We used the differential correlation coefficients to model the prognosis of breast cancer. Lasso regression is suitable for data analysis and model construction with many independent variables but a limited sample size (*Zhang et al., 2015*). In this study, we used a Lasso regression model to effectively reduce the dimensionality of large gene pairs and then identified the gene interactions related to the prognosis of breast cancer. Finally, a multivariate Cox proportional hazards regression analysis based on the gene interactions was carried out to predict the survival of patients with breast cancer (see Fig. 1B). A prognosis model was established and it also performed very well on an independent validation data.

## MATERIALS AND METHODS

### Datasets

In this article, the RNA sequencing (RNA-seq) data of 290 normal breast tissues was downloaded from the GTEx database (https://gtexportal.org/home/), and the RNA-seq data of 1,093 breast cancer samples was downloaded from the TCGA database (https://portal.gdc.cancer.gov/). A human protein–protein interaction network was from the STRING database version 11.0 (https://string-db.org/), and gene sets of all available186 KEGG pathways were downloaded from the GSEA/MSigDB database (http://software.broadinstitute.org/gsea/msigdb). In addition, the clinical information of the breast cancer patients was downloaded from the TCGA database, including TNM stage, prognosis survival time and other information. The 290 normal breast tissues were used as reference samples. The gene expression data sets of normal and cancer samples were both converted to the TPM form and contain 18,006 genes in total. The independent validation data of the prognosis model was from the GSE3494 set in GEO Datasets, which contains 251 expression profiles of breast tumors by array.

### Construction of sample-specific networks

In this study, gene–gene interactions with high confidence (comprehensive score >0.9) were selected from the STRING database, which include regulatory, physical and co-expression protein–protein interaction networks. Furthermore, the above gene–gene interactions with both genes in one of the 186 KEGG pathways were used as the background network (or template network), which contained 3,257 genes in total. The sample-specific network method aimed to calculate the difference of the gene co-expression when the single cancer sample was added to a bunch of normal samples. In short, the sample-specific networks to be constructed are actually networks with significant perturbation edges of gene co-expression.

In the following analysis, sample-specific networks for breast cancer samples were constructed based on gene expression profiles by using the method introduced in reference (*Liu et al., 2016*) (see Fig. 1A). First, using the gene expression data of $n$ reference samples, namely all the normal breast tissues data, the reference network can be constructed by calculating the correlation coefficient $PCC_n$ (the Pearson correlation coefficient (PCC)) of the gene pairs connected in the background network. The weights of the edges in the reference network are the PCC of the corresponding gene pairs. Then, the expression data of a single breast cancer sample was added to the reference samples, and the perturbed network of the single sample was constructed by calculating the new correlation coefficient $PCC_{n+1}$ of the gene pairs in the background network. For the single breast cancer sample, the differential correlation coefficients of each edge between the perturbed network and the reference network were calculated as: $\Delta PCC_n = PCC_{n+1} - PCC_n$, which called differential network for the sample. In reference *Liu et al. (2016)* have proved that $\Delta PCC_n$ follows a normal distribution with a mean value of 0 and a variance of $\dfrac{1 - PCC_n^2}{n - 1}$ when $n$ is large enough. The significance level of each $\Delta PCC$ was determined by the Z-test.

The statistical $Z$-value is calculated as follows with the null hypothesis that $\Delta\mathrm{PCC}_n$ is equal to 0:

$$Z = \frac{\Delta\mathrm{PCC}_n}{\left(1 - \mathrm{PCC}_n^2\right)/(n-1)}$$

Then, we can obtain the $P$-value for each gene pair from the $Z$-value. Gene pairs (or edges) were considered statistically significant if their $P$-values < 0.01. All significant edges constitute the sample-specific network. Thus, adding the expression data of 1,093 breast cancer samples to the reference samples one at a time, we finally constructed 1,093 sample-specific networks.

## Identification of stage/subtype-related gene–gene interaction networks

Only the gene pairs that are perturbed significantly in the most breast cancer samples are considered to be related to breast cancer. Then, the edges that are perturbed significantly in more than 90% of the samples by the binomial right-sided test ($P$-value < 0.05) constitute a gene–gene interaction network related to breast cancer. Specifically, we firstly divided the above breast cancer samples into different stages or subtypes based on TNM staging and PAM50 subtype system, secondly selected the perturbed significantly edges in more than 90% samples of different stages or subtypes, and then the stage/subtype gene–gene interaction networks were constructed.

A slight change in the expression of high-degree genes in the network may cause disturbances of the entire network. Thus, these genes with high degree are considered to be the key genes for the onset and development of breast cancer. We selected genes with degrees >5 in the identified breast cancer-related network for the subsequent enrichment analysis. Furthermore, we also identified the key genes related to each TNM stage and PAM50 subtype with the same method. Here, because of the small number of Stage V samples, Stage V was combined with Stage IV.

## Pathway enrichment analysis

For the pathway enrichment analysis, we used the hypergeometric test as follows:

$$p(m, M, N, n) = 1 - \sum_{i=0}^{m-1} \frac{\binom{M}{i}\binom{N-M}{n-i}}{\binom{N}{n}}$$

where $N$ is the total number of genes in the background network, $M$ represents the number of key genes related to breast cancer (or a stage or subtype of breast cancer), $n$ accounts for the number of genes in a pathway, and $m$ represents the number of genes that both in the pathway and in key genes related to breast cancer (or a stage or subtype of breast cancer). Then, the pathway with $P$-value < 0.05 was considered as significantly enriched in the breast cancer (or a stage or subtype of breast cancer) samples. Otherwise, we regarded that the pathway is not enriched in the corresponding group.

## Survival analysis by the Cox regression model

Different from the usual survival analysis based on gene expression, the perturbation of gene co-expression ΔPCC (i.e., gene pairs or edges) was used to survival analysis. According to the clinical data of patients with breast cancer, we utilized the "survival" package and "survminer" package in R/Bioconductor to establish a univariate Cox proportional hazards regression model by setting patients' survival conditions (survival time and survival status) as the dependent variables and the ΔPCC of gene pairs in the differential network for each breast cancer samples as the covariates. Gene pairs with $P$-values < 0.05 were considered to be related to the prognosis of breast cancer (*Cheng, 2018*).

A large number of covariates may cause overfitting in establishing a multivariate Cox proportional hazards regression model; thus, using the least absolute shrinkage and selection operator (LASSO), we further selected the key gene pairs from these significant ones obtained by the univariate Cox proportional hazards analysis. LASSO is a common method used in high-dimensional data regression, which can select prognosis-related gene pairs of breast cancer by shrinking regression coefficients. The tuning parameter (λ) with the smallest mean-square error was selected by four-fold cross-validation to establish an optimal LASSO regression model. Then, the coefficients of most gene pairs reduced to zero, and a small number of gene pairs with nonzero coefficients were considered to be closely correlated with the prognosis of breast cancer.

LASSO Cox analysis was performed by using the "glmnet" package in R. Then, the risk score for each sample was calculated by the LASSO Cox regression model. According to the median risk score, breast cancer patients were divided into two groups (a high-risk group and a low-risk group). In addition, 234 breast tumors with relapse free survival information in the validation data set were analyzed by using the above sample-specific network method, and risk scores were calculated by the Cox regression model based on 1,093 samples in TCGA. Then the validation samples were also divided into two groups in the same way. Finally, the corresponding Kaplan–Meier survival curves were plotted by using the packages "survminer" and "survival" in R.

# RESULTS

## Breast cancer-related gene–gene interaction networks

The background network consisted of 46,916 edges and 3,237 genes. In addition, 2,190 gene pairs were identified as significantly related to breast cancer, which constituted the gene–gene interaction network related to breast cancer (including 915 genes in total). We use the Cytoscape software to visualize the breast cancer-related network (see Fig. 2).

Genes with degrees >5 in the breast cancer-related gene–gene interaction network (198 in total which are shown in Table S1). Among them, some genes with higher degrees (>20) have been shown to be related to breast cancer. For example, CCNB1 has strong power to predict the survival of breast cancer patients with the phenotype of ER positive (*Ding et al., 2014*). The overexpression of GRB2 has been demonstrated to be significantly associated with the occurrence and poor prognosis of breast cancer (*Zhang et al., 2016*). PCNA has been proven to be a marker of proliferation in the diagnosis of breast

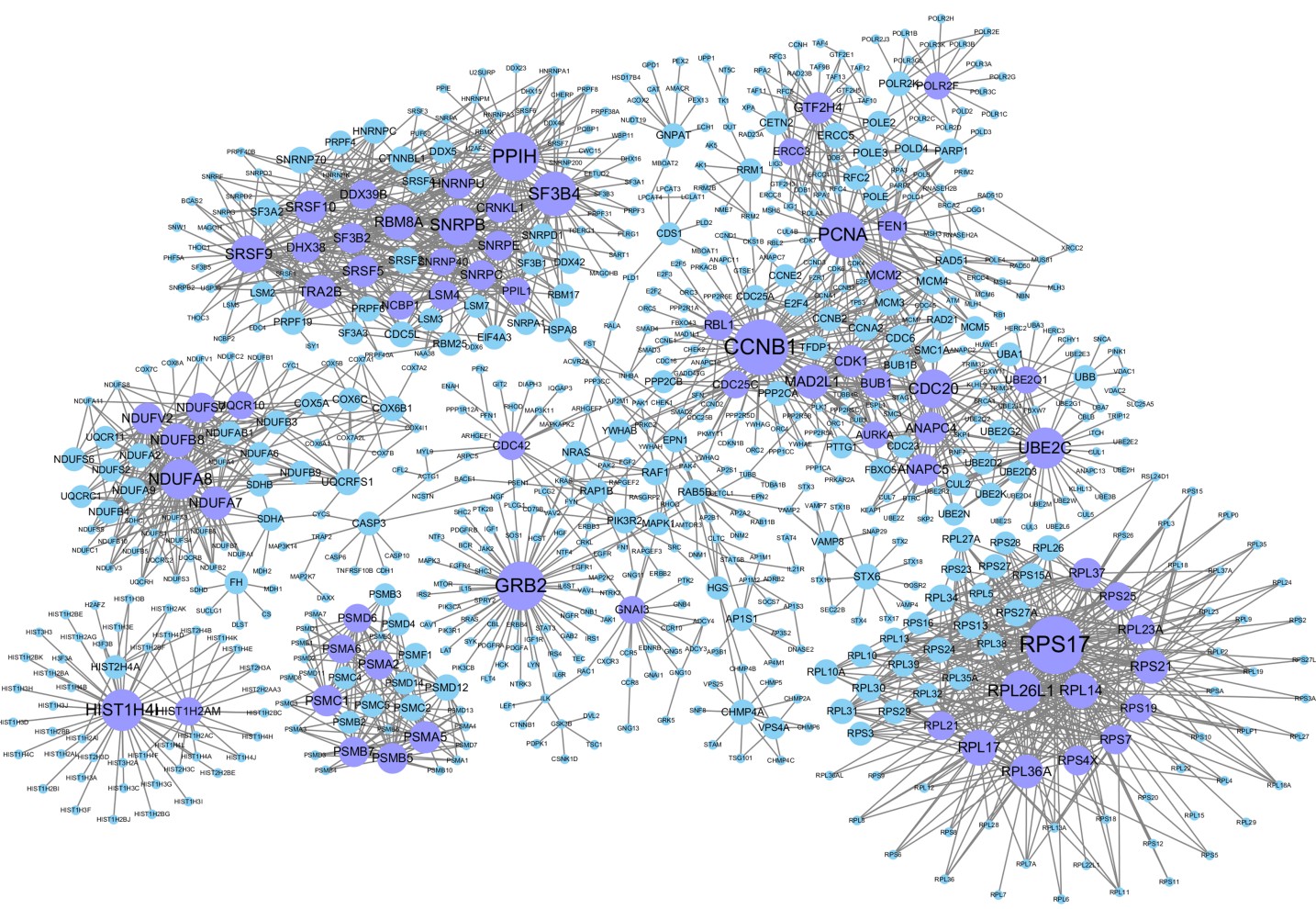

**Figure 2 Gene–gene interaction networks related to breast cancer.** Nodes in these networks stand for genes, and the size of the nodes corresponds to the degree of the genes in the network. The purple nodes represent the genes with degrees ≥15, and the blue ones are the genes with degrees <15.

cancer (*Juríková et al., 2016*), SF3B4 has been shown to be a tumor suppressor, and somatic inactivating mutations occasionally occur in breast cancer (*Denu & Burkard, 2017*). UBE2C may promote the development of breast cancer (*Mo et al., 2017*). High Cdc20 and securin immune expression are associated with extremely poor outcomes in breast cancer patients (*Karra et al., 2014*), and overexpression of RPL17 affects breast cancer-associated brain metastases (*Yuan, Wang & Cheng, 2018*). MAD2L1 may have great effect on breast cancer progression, and its expression might help to predict breast cancer prognosis (*Wang et al., 2015*). The high expression of TRA2B is closely related to the cancer cell survival and therapeutic sensitivity of breast cancer (*Best et al., 2013*). GTF2H4 has been identified to be related to the survival risk of breast cancer (*Ge et al., 2019*).

## Stage-related gene–gene interaction networks

The results of the four stage-related gene–gene interaction networks are shown in Figs. S1A–S1D. And the top 10 genes with the highest degrees in these four networks are

displayed in Figs. S1E–S1H. There are obvious similarities and differences among the four stage-related gene interaction networks. There are 81 key genes shared by all stages (see Table S3), among which RPL17, CCNB1, and SF3B4 are genes that are highly (with degrees >25) related to breast cancer. Stage I has 5 specific genes: PSMC5, SDHB, RPL11, SDHA, and RPL13. Stage II has 3 specific genes: STX6, CCNA2, and CDC25C. Stage III has 4 specific genes: NDUFA6, EPN1, SF3A3, and LSM7. Stage IV has the largest number of specific genes, with a total of 38, among which CDC42, LSM2, NDUFS6, and CDC25A are strongly associated with it. And these stage-specific key genes are shown in Table S4.

## Subtype-related gene–gene interaction networks

The results of the four subtype-related gene interaction networks are shown in Figs. S2A–S2D. And the top 10 genes with the highest degrees in these four networks are displayed in Figs. S2E–S2H. The four subtype-related networks share similar and different characteristics. There are 34 key genes shared by the four subtypes (see Table S7). Among them, RPL17 and CCNB1 have higher degrees. The Luminal A subtype has 11 specific genes, including RPL23A, RPL10, and PRPF6, which are greatly related to it with higher degrees. The Luminal B subtype has 17 specific genes, including COX6C, EGFR, and CLTC, which are related to it with higher degrees. The Her2 subtype has 3 specific genes, NDUFA6, CCR8, and CASP3. The Basal-like subtype has the largest number of specific genes, 17 in total, including LSM2, DDX5, SF3A3, and MAGOH, with higher degrees. And these subtype-specific key genes are shown in Table S8.

## Pathways enriched in breast cancer patients

There were 41 pathways (see Table S2) enriched in the breast cancer samples according to the pathway enrichment analysis, including some immune-related pathways, such as the Toll-like receptor signaling pathway, antigen processing and presentation, complement and coagulation cascades, the RIG-I-like receptor signaling pathway, and the cytosolic DNA-sensing pathway. Some important signal transduction and signal molecular interaction pathways were also included, such as the MAPK signaling pathway, Wnt signaling pathway, cytokine-cytokine receptor interaction, and ECM–receptor interaction pathways. Breast cancer is closely related to endocrine disorders (Sakoda et al., 2008), two endocrine-related pathways, adipocytokine signaling pathway, and PPAR signaling pathway, have also been identified as being related to breast cancer. In addition, some metabolic pathways, especially lipid metabolism pathways, have also been identified as being associated with breast cancer (Merdad et al., 2015), such as the steroid hormone biosynthesis, arachidonic acid metabolism, arginine and proline metabolism pathway, and glycerolipid metabolism. Additionally, pathways in cancer was also enriched. The enrichment results are shown in Fig. 3A.

Most of these pathways have been documented to be related to breast cancer. For example, the dysregulation of the steroid hormone biosynthesis pathway may affect steroid hormone levels and may thus be related to the susceptibility to breast cancer (Sakoda et al., 2008). The PPAR signaling pathway may play an important role in the

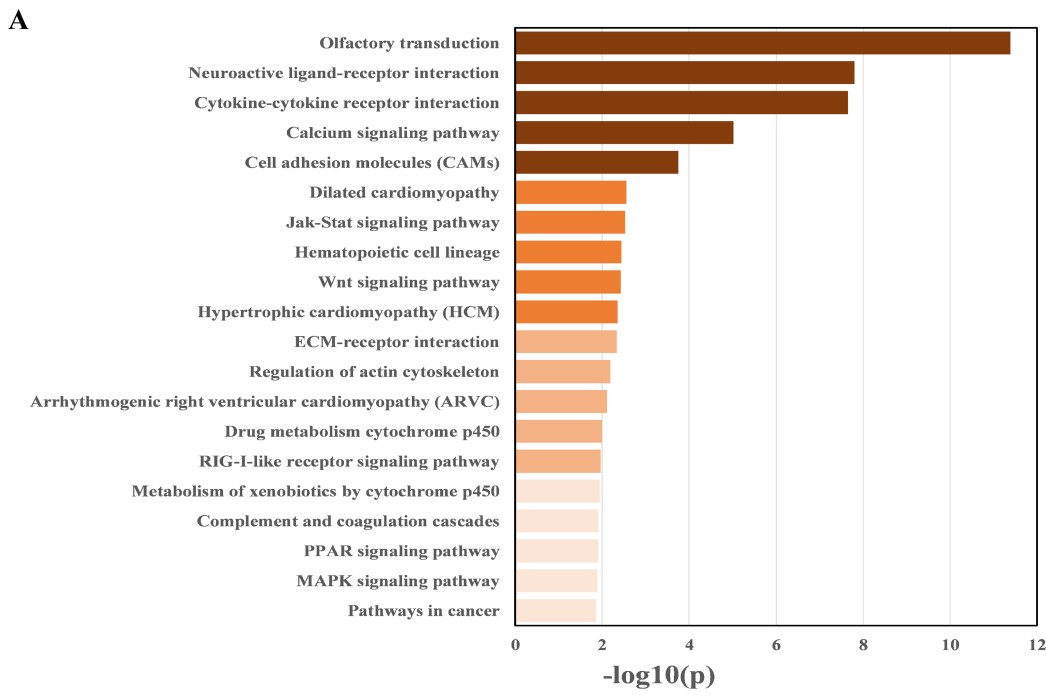

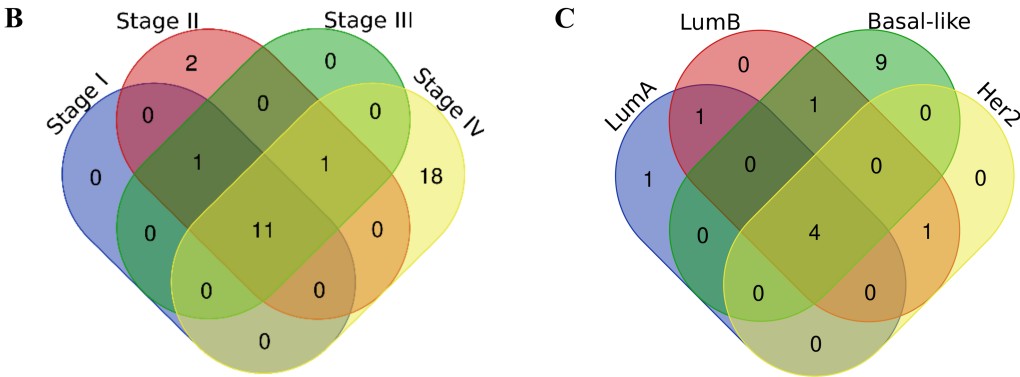

**Figure 3 Pathways enriched in breast cancer, as well as different stages and subtypes of it.** (A) KEGG pathways enriched in breast cancer samples, ranked by −log10(p). (B) Overlap and difference of the enriched pathways in the four breast cancer stages. There are 11 commonly enriched pathways in the four stages. The number of Stage IV-specific pathways was 18. (C) Overlap and difference of the enriched pathways in the four PAM50 subtypes. There are four commonly enriched pathways in the four PAM50 subtypes. The number of Basal-like specific pathways is 9.

neoadjuvant chemotherapy response of breast cancer (*Chen et al., 2012*). Mounting preclinical evidence supports targeting the MAPK signaling pathway in the triple negative breast cancer (TNBC) (*Giltnane & Balko, 2014*). AMPK activators inhibit breast cancer cell proliferation by inhibiting DVL3-promoted Wnt/β-catenin signaling pathway activity (*Zou et al., 2017*). Toll-like receptors may play dual roles in human cancers (*Khademalhosseini & Arababadi, 2019*). The co-activation of the Hedgehog and Wnt signaling pathways is a poor prognostic marker in TNBC (*Bhateja et al., 2019*). Prl-3 is closely related to cell migration and invasion in TNBC (*Gari et al., 2016*). The YHD

inhibition of 4T1 breast tumor growth may be related to the negative regulation of the JAK/STAT3 pathway by repressing the expression of IL-6 and TGF-β (*Mao, Feng & Gong, 2018*).

### Stage-related pathways

The overlapping of pathways enriched in the four TNM stages are shown in Fig. 3B. The proportion of enriched pathways shared by the four stages (see Table S5) is relatively high, including the Wnt signaling pathway, MAPK signaling pathway, regulation of actin cytoskeleton, calcium signaling pathway, pathways in cancer, and cell adhesion molecules, which have been shown to have a high correlation with breast cancer (*Giltnane & Balko, 2014*; *Zou et al., 2017*; *Kazazian et al., 2017*; *Woltmann et al., 2014*; *Saadatmand et al., 2013*). The pathways enriched in different stages are slightly different, especially Stage IV of breast cancer, which has 18 specific enriched pathways, among which the PPAR signaling pathway, ECM-receptor interaction, tight junction, TGF-beta signaling pathway, NOD-like receptor signaling pathway, and other signaling pathways are mostly related to the metastases of breast cancer (*Chen et al., 2012*; *Bao et al., 2019*; *Yang et al., 2019*; *Tang et al., 2017*; *Peng et al., 2016*).

As we expected, Stage IV was specifically enriched the most pathways (18 in total, see Table S6) different from other stages. This result is probably because Stage IV breast cancer patients are the most serious, and their cancer cells are likely to have deteriorated and metastasized. Therefore, the disruption of the biological system balance of breast cancer patients at this stage is larger than that of other stages. Thus, the specific enriched pathways of Stage IV are correspondingly more.

### Subtype-related pathways

The overlap and difference of the enriched pathways in the four PAM50 subtypes are shown in Fig. 3C. There are slight differences in the subtype-related pathways. There are 4 enriched pathways shared by the four subtypes (see Table S9) including the cytokine–cytokine receptor interaction. As a special subtype of breast cancer, the Basal-like subtype (or TNBC) is characterized by high histological differentiation, a high risk of metastasis, a high recurrence rate, and a low survival rate. Probably due to the higher risk of Basal-like subtype, there are 9 specific pathways enriched in it, including the leukocyte transendothelial migration and chemokine signaling pathway. The subtype-specific enriched pathways are shown in Table S10.

### Prognosis-related gene pairs

A total of 5,652 gene pairs significantly related to the survival and prognosis of breast cancer were found by the univariate Cox proportional hazards model. In addition, 272 gene pairs were further identified by Lasso regression (see Fig. S3). A multivariate Cox proportional hazards regression model with these gene pairs as independent variables was constructed as follows: Score = 206.3 ∗ (ENO, PGK2) + 35.9 ∗ (EN0, PKLR) + 4.1 ∗ (EBP, HSD17B7) + 5.5 ∗ (CYP1B, HSD17B1) − 3.4 ∗ (NDUFB2, NDUFB4) − 0.6 ∗ (ATP6V1A, ATP6V1B1) + ….

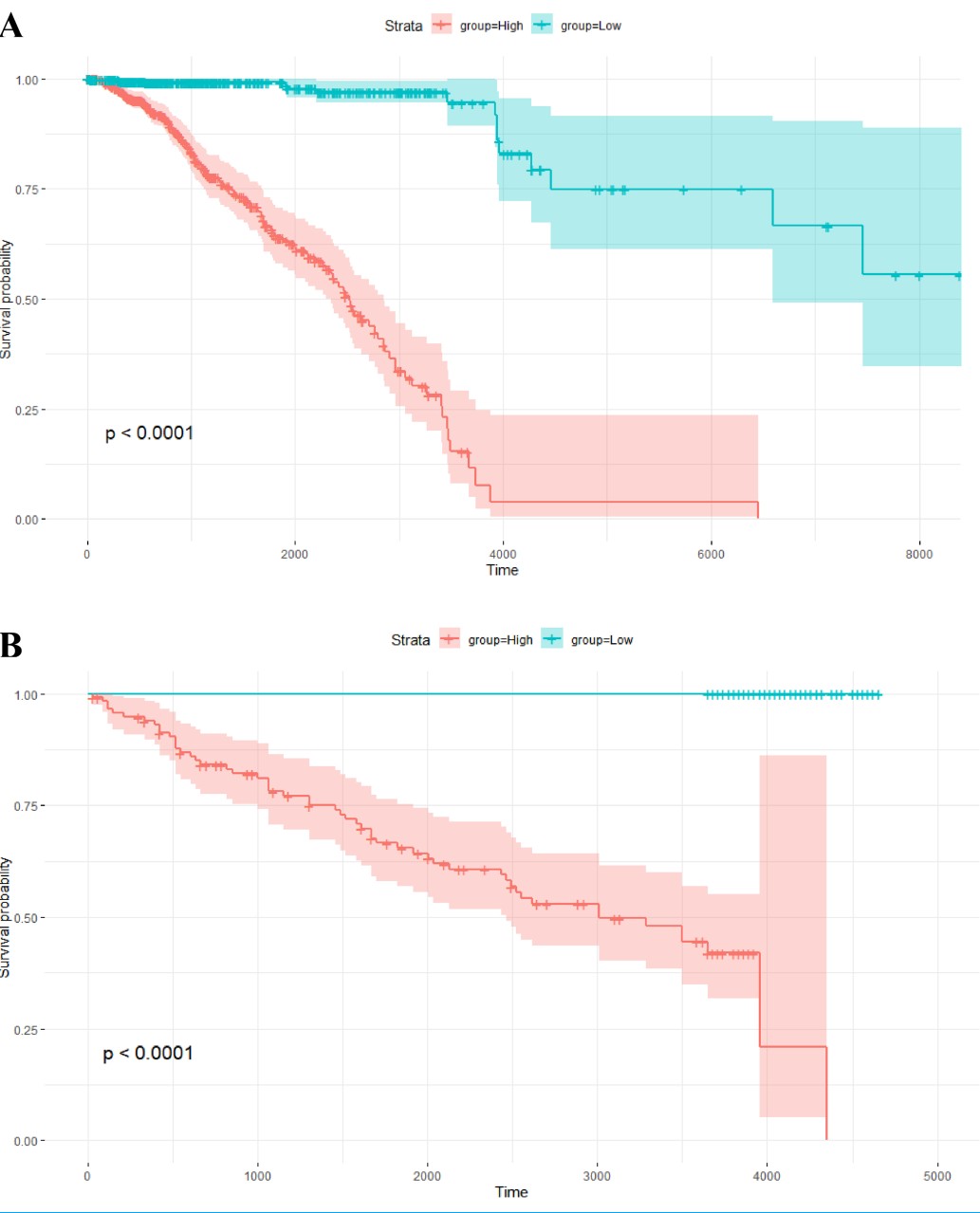

**Figure 4 Kaplan–Meier survival analysis.** (A) Kaplan–Meier survival plots for two different groups of breast cancer patients in TCGA. The $X$ axis is survival days. The $Y$ axis is overall survival rate. (B) Kaplan–Meier survival plots for two different groups of breast tumors in the independent validation data set. The $X$ axis is relapse free survival time (days). The $Y$ axis is relapse free survival rate.

The risk scores of the 1,093 breast cancer patients in TCGA were calculated by this model. The median of the risk scores divided all patients into two groups. The corresponding Kaplan–Meier survival curve is shown in Fig. 4A. Of note, survival analysis indicates that overall survival probability of patients with high risk scores is significantly lower than that with low risk scores ($P$-value < 0.0001).

In addition, the risk scores of the 234 breast tumors in the validation data set were also calculated by the above model with 264 gene pairs (8 gene pairs were omitted since these genes were not included in the expression profile of the validation data). In the same way, there are two groups with different scores. The relapse free survival probabilities of the two groups are significantly different (*P*-value < 0.0001), and the relapse free survival status of tumors in the low score group are all "alive" (see Fig. 4B). This result indicates that the prognosis model based on gene pairs can well predict the survival time of breast tumors in the independent validation data set.

## DISCUSSION

At present, research on cancer pathology is limited to gene expression and mutation information. However, the model of one gene to one disease is no longer suitable for the study of complex diseases. In fact, genes do not exist in isolation but participate in some complex biological networks, such as gene–gene interaction networks. Gene mutations or surroundings changes often affect the balance of gene interaction networks and the perturbation of the networks then affect the onset and development of complex diseases. Studies have shown that some genetic elements of breast cancer are related to nearby gene expression, such as some repetitive DNA in ER+/HER2-breast cancer and transposable elements (*Yandım & Karakülah, 2019*; *Karakülah et al., 2019*). Therefore, network analysis can provide a more comprehensive and systematic point of view, to better understand the human disease onset and development mechanism.

Based on personalized medicine, Precision Medicine is a new medical concept and medical model, which needs to grasp the specific characteristic of different cancer samples accurately. The analysis of the biological network disturbance for each cancer patient conforms to the concept of precision medicine. In addition, the personalized medical treatment of breast cancer is in a relatively slow development stage.

In this paper, sample-specific networks of breast cancer samples were established to explore the gene–gene interaction networks related to the TNM stages and PAM50 subtypes of breast cancer. Then, the pathways related to breast cancer were identified by hypergeometric test. Through the same method, we also obtained the stage-related pathways and subtype-related pathways. Finally, the edge biomarkers (gene pairs) that are closely related to the prognosis of breast cancer were determined by using the LASSO regression model, and then a more stable prognostic analysis model was established by using these biomarkers. Our results indicate that the prognosis model has the robust and strong generalization capability, and it can be used in different gene expression data sets.

Many studies have shown that network-based methods are more robust and effective than single-gene-based methods, such as SWIM and WGCNA (*Paci et al., 2017*; *Langfelder & Horvath, 2008*). SWIM is a tool able to extract from complex correlation networks the so-called "switch genes" that could be associated to the transition from physiological to a pathological condition. The WGCNA method plans to exploit the correlation patterns among genes. The advantages of network-based methods have been well documented and accepted in the analysis of noisy high-throughput data.

Different from the usual network-based method, we made better use of a prior background network to explore the sample-specific networks. And the sample-specific networks are actually networks with significant perturbation edges of gene co-expression in our study, which is really very different from WGCNA. This study helps us to better understand the heterogeneity and mechanism of breast cancer from an individual-level perspective. Precision medicine advocates the development of individualized treatment according to the unique features of patients. Therefore, identifying the unique pathogeny embedded in each patient is important to develop a treatment strategy for each patient. Our sample-specific network analysis of breast cancer will promote the development of precision medicine.

## CONCLUSIONS

In this article, the sample-specific network of each breast cancer sample was constructed based on network analysis, and further breast cancer (subtype/stage)-related gene–gene interaction networks were identified. The edge biomarkers (gene pairs) related to the prognosis of breast cancer were also identified and a risk prediction model was established based on these edge biomarkers finally.

This study develops an individualized network analysis for each patient which would promote a new train of thought and method for the precision medicine. This whole process of sample-specific network analysis using co-expression can also be used to analyze other cancers. However, the co-expression perturbation which used to construct sample-specific network, does not roundly measure the changes of gene interactions. So, we will consider further designing a method which can characterize the perturbation of gene interactions comprehensively. In addition, how to obtain subtype-specific networks (or stage-specific networks) from sample-specific networks based on network structure is still a problem worth considering.

## ACKNOWLEDGEMENTS

We are very grateful to Prof. Zhaohui Qin for his useful advice.

### Funding

This work was supported by the China Postdoctoral Science Foundation (No. 2019M651658), and the National college students' innovation training program (No. 201910307068Z). The funders had no role in study design, data collection and analysis, decision to publish, or preparation of the manuscript.

### Grant Disclosures

The following grant information was disclosed by the authors:
China Postdoctoral Science Foundation: 2019M651658.
National College Students' Innovation Training Program: 201910307068Z.

## Competing Interests

The authors declare that they have no competing interests.

## Author Contributions

- Ke Zhu analyzed the data, authored or reviewed drafts of the paper, and approved the final draft.
- Cong Pian analyzed the data, authored or reviewed drafts of the paper, and approved the final draft.
- Qiong Xiang performed the experiments, analyzed the data, authored or reviewed drafts of the paper, and approved the final draft.
- Xin Liu performed the experiments, analyzed the data, prepared figures and/or tables, and approved the final draft.
- Yuanyuan Chen conceived and designed the experiments, performed the experiments, prepared figures and/or tables, authored or reviewed drafts of the paper, and approved the final draft.

## Data Availability

The data is available at the:

1. TCGA database (search terms: TCGA-BRCA, Transcriptome Profiling, Gene Expression Quantification, HTSeq-FPKM).

2. GTEx database (https://gtexportal.org/home/, search terms: RNA-seq Data, Genes TPMS, Breast).

3. STRING database version 11.0 (https://string-db.org/, search terms: Download, Human sapiens, interaction data, protein network data (scored links between proteins)).

4. GSEA/MSigDB database, gene set: CP:KEGG (https://www.gsea-msigdb.org/gsea/msigdb/genesets.jsp?collection=CP:KEGG).

5. NCBI GEO: GSE3494.

Link CP:KEGG to "https://www.gsea-msigdb.org/gsea/msigdb/genesets.jsp?collection=CP:KEGG".

## Supplemental Information

Supplemental information for this article can be found online at http://dx.doi.org/10.7717/peerj.9161#supplemental-information.

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
