# Peer review of "Personalized analysis of breast cancer using sample-specific networks"

_PeerJ, doi:10.7717/peerj.9161_

## Round 0.1 · original submission · Major Revisions

I would like to invite the authors to response the reviewers' comments. Additionally, my own assessment as academic editor, I recommend the inclusion of following citations, which indicates the importance of other genetic elements in breast cancer: PMID: 31536958 and PMID: 31824778

Reviewer 1 ·

Basic reporting

- The authors build individual-level gene expression networks in order to find breast cancer-specific interactions and pathways related to tumor staging and subtypes. The manuscript addresses a relevant topic to the scientific community and the study is scientifically sound, but some improvements and clarifications need to be performed to the presentation of the study. In particular, the authors perform a standard bioinformatics analysis, where the novelty is not properly highlighted. I think that it would be very interesting to conduct a network analysis to find communities in the PPI network and investigate the behavior of modules/genes that are co-expressed, co-localized in the PPI network and that could share functionality, specifically related also to the topic under study, that is breast cancer and its heterogeneity. Indeed, there is a new and cutting-edge field of medical research, called network medicine, whose basic idea is that human diseases are rarely caused by single molecular determinant, but more likely influenced by a network of interacting molecular determinants with the propensity to cluster together in the human interactome.
- The authors addressed a relevant and interesting topic that fits well a new and cutting-edge field of network medicine (see for example reference [1-3]). This aspect should be discussed and highlighted in the Introduction or in the Discussion (example at line 292).

References
1. Albert-László Barabási et al. “Network medicine: a network-based approach to human disease” Nature Reviews Genetics volume 12, pages 56–68 (2011) https://doi.org/10.1038/nrg2918
2. Conte et al. (2019) “A paradigm shift in medicine: a comprehensive review of network-based approaches”. Biochimica et Biophysica Acta (BBA)-Gene Regulatory Mechanisms. Article num: 194416 https://doi.org/10.1016/j.bbagrm.2019.194416
3. Fiscon et al. (2018). “Network-based approaches to explore complex biological systems towards network medicine”. MDPI Genes, 9(9), 437.https://doi.org/10.3390/genes9090437

Experimental design

- In the framework of network medicine, many network-based methods have been developed to aid in understanding molecular-level underpinnings complex human diseases. In particular, for what concerns the building and analysis of gene co-expression networks (see reference [3]), the two widespread algorithms are SWIM (see reference [4]) and WGCNA (see reference [5]). SWIM is a software able to extract from complex correlation networks the so-called “switch genes” that could be associated to the transition from physiological to a pathological condition. Among its several applications, SWIM was applied to study a large panel of gene expression data of tumors from TCGA, including breast cancer. WGCNA is a software exploited for describing the correlation patterns among genes and that can be used for finding clusters (modules) of highly correlated genes, for summarizing such clusters using the module eigengene or an intramodular hub gene, for relating modules to one another and to external sample traits (using eigengene network methodology), and for calculating module membership measures. Both algorithms (see references [4-5]) I think should be mentioned in the Discussion, for example at line 311-312 when the authors claim: “Many studies have shown that network-based methods are more robust and effective than single-gene-based methods”
- From all the considerations explained above, I think that could be very interesting and thus strongly recommend for the authors to apply WGCNA software to study and deeply analyze their correlation networks. By using WGCNA the author will be able to find network modules and also identify which are those modules that are more correlated (and thus strongly associated) for example with the tumor staging, or also to other clinical traits of interest.
- For what concerns pathway the pathway enrichment analysis, did the authors make use of some tool/R package as enrichR? If yes, it should be specified.

References
4. Paci, P. et al. (2017) "SWIM: a computational tool to unveiling crucial nodes in complex biological networks." Scientific reports 7: 44797. https://www.nature.com/articles/srep44797
5. Langfelder, P., Horvath, S. (2008) WGCNA: an R package for weighted correlation network analysis.BMC Bioinformatics 9, 559. https://doi.org/10.1186/1471-2105-9-559

Validity of the findings

As mentioned in the “experimental design” section, the obtained results and their interpretation could be extended by exploiting the WGCNA analysis. For example, it could be also really interesting to check if the key genes identified by the authors for breast stages and for subtypes form unique modules in the human interactome.

Additional comments

- Figures quality is too low and should be improved, in particular:
o Figure 1: panel A bottom: “significanc” should be “significance; panel B bottom: plot of survival analysis is not visible.
o Figure 2: network edges could be colored by p-value.
o Figure S1: networks are barely readable, it is not informative in this form.
- In general, the authors should check the English language and the form of the entire paper avoiding careless errors, missing commas, and confusing sentences that should be reworded in order to make them clearer. In the following, I have just reported some examples.
- Check all the commas. Add a comma before “and” whenever more than two elements have been listed (E.g. line 51, line 250, etc…)
- Line 34: missing space between 24% and among
- Line 42, 83: missing references
- Line 314-315: please reformulate the sentence, it is not clear and appears a truncated phrase.

Reviewer 2 ·

Basic reporting

Adopting a sample-specific networks, the authors further investigated the breast cancer specific networks in various aspects including cancer stage, subtype, pathway and survival. In general, the study was designed reasonable and the findings were valuable to the research field.

Experimental design

no comment

Validity of the findings

no comment

Additional comments

Here are still some spaces for improving the article:
1. The text was not clear enough,
a. First of all, the term individual-level network was not quite meaningful without clear explanation in this manuscript. The reference [7] was the paper brings up the network method. The title was “Personalized characterization of diseases using sample-specific networks” which is different from the citated reference. Please double check.
b. Are the 186 KEGG pathways involved in this paper breast cancer related? It should be described clearly. Line 96.
c. The description of the network construction was not clear. For example: “significant perturbations of co-expression”, line 110. What are the n reference samples? Are they the normal samples? Line 113.
d. I suggest putting more details into section “Identification of stage/subtype-related gene-gene interaction networks”. The authors mentioned some important genes were selected and used for the analysis. However, the details for identifying stage/subtype-related specific networks are needed.
e. Similarly, I suggest adding some more details in “Survival analysis by the Cox regression model” section. How the individual-level networks used for this analysis.
2. The validation,
GSE3494 was mentioned to be used as validation data set. From line 171 to line 173, only one sentence here describing the validation process.
The question is, did you apply the Cox model or the network method on this data set, which means did you fit a specific Cox model over the GSE3494 data or used the Cox model based on 1093 samples.
From the text I assume you fitted a new model. In this case, I think this data set is not used for validation. It is just a second data set for the survival analysis because there is no comparison can be taken between them.
3. The other parts of the manuscript should be revised correspondingly.

---

## Round 0.2 · accepted · Accept

The authors successfully implemented the criticisms and comments raised by the reviewers during the revision period. Therefore, the manuscript can be accepted in its current form.

Reviewer 1 ·

Basic reporting

The author carefully addressed all my comments, thus I suggest to accept the manuscript

Experimental design

The author carefully addressed all my comments, thus I suggest to accept the manuscript

Validity of the findings

The author carefully addressed all my comments, thus I suggest to accept the manuscript

Additional comments

The author carefully addressed all my comments, thus I suggest to accept the manuscript